# Development of a blood test for uterine sarcoma—Diagnosis and monitoring (DOORS-D and DOORS-M) studies

Anna Casey[1,2], Rebecca Allsopp[1], Jacqui Shaw[1], Caroline Cowley[1], Elizabeth Stannard[2], Yvette Griffin[2], Indrajeet Das[2], Marc Wadsley[1], Natalie Darko[3], Esther L. Moss[1,2]*

1 Leicester Cancer Research Centre, College of Life Sciences, University of Leicester, Leicester, United Kingdom, 2 University Hospitals of Leicester NHS Trust, Infirmary Square, Leicester, United Kingdom, 3 College of Life Sciences, University of Leicester, Leicester, United Kingdom

* em321@leicester.ac.uk

## Abstract

Uterine sarcomas can be difficult to differentiate from uterine fibroids due to many shared symptoms and imaging features, which can result in delayed or missed diagnosis, or over treatment. The 'Development of a blood test for uterine Sarcoma – Diagnosis' (DOORS-D) (ISRCTN14800787) and 'Development of a blood test for uterine Sarcoma – Monitoring' (DOORS-M) (ISRCTN14174468) studies aim to explore the role for circulating tumour DNA (ctDNA) to diagnose and to monitor uterine sarcomas. DOORS-D will recruit patients who have a suspected uterine sarcoma or large fibroid and are due to undergo surgery (hysterectomy or myomectomy) for a blood sample prior to surgery whereas DOORS-M will recruit patients who have been diagnosed with a uterine sarcoma in the previous 10 years for longitudinal blood sampling every 3–6 months over the course of the study. Information will be collated on patient characteristics and symptoms, tumour characteristics and diagnostic imaging, with representative images selected and analysed using large language models. Analysis of genomic/methylation profile of ctDNA samples collected from DOORS-M will be used to design a ctDNA-based 'test'. Analysis of the samples collected from the participants recruited to the DOORS-D study will enable the accuracy of the 'test' to differentiate uterine sarcomas from fibroids to be determined. In addition, the opinions of patients with a suspected or confirmed sarcoma will be explored through semi-structured qualitative interviews. Purposeful recruitment strategy will ensure that the experiences of women from diverse socioeconomic, cultural and ethnic backgrounds are included. The results of the studies will be shared through conference presentations and peer-reviewed publications.

**Data availability statement:** No datasets were generated or analysed during the current study. All relevant data from this study will be made available upon study completion.

**Funding:** EM, JS and DD were the recipients of the grant from The Eve Appeal, the gynaecological cancers charity (EVE 0042) https://eveappeal.org.uk/. The funder has not had a role in the design of the study and will not be involved in the conduct, analysis, or reporting of trial. The University of Leicester is the study sponsor. The research team (EM/JS and DD) are employees of the sponsor and have designed of the study and will conduct, analyse, and report of outcomes. This study will be delivered through the National Institute for Health and Care Research (NIHR) Leicester Biomedical Research Centre. The views expressed are those of the author(s) and not necessarily those of The Eve Appeal, the NIHR Leicester BRC, the NIHR or the Department of Health and Social Care.

**Competing interests:** EM/DD have received research funding from Sarcoma UK for an unrelated sarcoma-focused research study. This does not alter our adherence to PLOS ONE policies on sharing data and materials.

## Introduction

Uterine sarcomas are rare mesenchymal tumours that account for 3% of all uterine cancers and are often associated with a poor long-term prognosis [1]. Uterine sarcomas typically arise from myometrial smooth muscle or endometrial stroma, with leiomyosarcomas (LMS) being the most common subtype accounting for 60–70% of cases, followed by endometrial stromal sarcomas (ESS) at 10–20% [2]. Their myometrial, rather than endometrial, location makes diagnosis more challenging since biopsy of the endometrium is only able to detect 35% of LMS and 25% of ESS [3]. The other key diagnostic challenge is the difficulty differentiating a uterine sarcoma from uterine fibroids which are very common benign uterine growths affecting 40–80% of women [4], due to shared symptomatology [5]. Fibroids especially impact women aged 40–60 years, who have a 4–11 times greater risk than younger women (20–30 years) or older women (>60 years) [6]. Women of African ancestry also experience a higher incidence and prevalence of uterine fibroids compared to women from other races or ethnic groups, 30.6/1000 women-years compared to 8.9/1000 for White women [7].

Accurately differentiating between a uterine fibroid and a sarcoma is of paramount importance, not only to expedite oncology management of patients with a sarcoma, but to avoid procedures that could cause result in cancer dissemination and thereby worsening patients' prognosis. Although the incidence of unexpected sarcoma at uterine fibroid surgery is low, 0.175 to 0.67% [5], it has been reported that 19% of unexpected sarcoma cases underwent a subtotal hysterectomy or open myomectomy [8]. Intraoperative pathology assessment can help with guiding patient management however, this is can be associated with diagnostic challenges [9], and appears to be less concordant with final histology compared to assessment of other gynaecological malignancies [10]. Laparoscopic myomectomy with morcellation in particular, is associated with a significantly lower 5-year recurrence-free survival rate compared to total hysterectomy for unexpected uterine sarcoma, 24.1% compared to 43.6%, with greater rates of peritoneal dissemination reported, 22.2% versus 0.0% [11]. Consequentially, the United States Food and Drug Administration (FDA) have advised morcellation within retrieval bags to prevent dissemination [12,13].

At present there is no specific blood test that can be used pre-operatively to differentiate between uterine sarcomas or fibroids. The tumour marker CA125 is not helpful [14]. LDH and its isoform LDH3 have been shown to be an independent predictor of LMS [15] although not validated for clinical use, and obesity can be a confounding factor increasing the false positive rate [16]. Other blood markers, such as C-reactive protein and absolute neutrophil count, have shown associations with sarcomas [17] however lack specificity. Imaging, in particular MRI, can be used to differentiate between uterine sarcomas and fibroids with a high level of diagnostic accuracy [18]. Features including peritoneal or lymph node metastases, high/intermediate on T2-weighted imaging, low apparent diffusion coefficient (ADC) and high intensity signal of diffusion weighted images were identified as having a strong association with uterine sarcomas [19]. Other features, such an irregular margin, haemorrhage and

enhancement, were determined as being an indeterminate association [19], and it is reported that there is a greater challenge when differentiating degenerating or complex fibroids from uterine sarcomas [20]. PET-CT scans have been shown to have a greater ability to distinguish fibroids from sarcomas, sensitivity of 80.8% and specificity of 100% [21], however, such imaging would typically only be performed in cases where there was a high suspicion of sarcoma, therefore is unlikely to impact the unexpected sarcoma diagnosis rate.

The genomic profile of uterine sarcomas reveals a mixed picture with common somatic mutations, including *TP53* (56%) and *RB1* (51%) but also somatic copy number alterations (SCNAs), with whole genome duplication (WGD) identified in half of high-grade sarcomas [22]. Uterine fibroids are monoclonal tumours although 40–50% are reported to have detectable chromosomal abnormalities, and a recent genome wide meta-analysis has identified 46 novel genes associated with fibroids [23]. Genomic analysis, for example, shallow whole genome sequencing (sWGS) shows potential to discriminate between sarcomas and fibroids, with a pilot study reporting that the percentage of the genome affected by SCNAs in plasma of six patients with fibroids was lower than the proportion of the genome showing SCNA in patients with sarcoma [24].

Liquid biopsies based on circulating tumour DNA (ctDNA) are rapidly becoming established as a minimally invasive tool for diagnosis, guiding treatment regimens and monitoring for recurrence [25]. The evidence to support a role in the monitoring and detection of endometrial cancer recurrence is increasing [26–29] with a higher diagnostic accuracy compared to the currently available tumour markers, CA125 and HE4 [30]. The potential of the 'liquid biopsy' to monitor uterine sarcomas has been explored in previous studies where ctDNA has shown promise as a marker both in the monitoring [31–33] and diagnosis [24] of uterine sarcomas, although other analytes including microRNAs have also been shown to have potential as biomarkers [34]. Leiomyosarcomas were the most common histological subtype investigated and there was significant heterogeneity in terms of both laboratory approaches and the reporting of these approaches. Given the small number of uterine sarcoma cases in the majority of the studies to date, the proportion of cases with detectable ctDNA is currently unknown, and there is no cost-effective test that could give a result in a short time frame to guide patient management, thereby reducing pre-operative diagnostic uncertainty.

### Hypothesis

ctDNA is an acceptable test that can be used in routine clinical practice to diagnose and/or monitor uterine sarcomas.

## Materials and methods

### Ethical approval and trial registration

In order to address the two potential roles for ctDNA, two studies have been designed: Development of a blood test for uterine Sarcoma – Diagnosis (DOORS-D) and Development of a blood test for uterine Sarcoma – Monitoring (DOORS-M). Both studies have received ethical approval from the South Central – Berkshire B Research Ethics Committee (25/SC/0107 and 25/SC/0108) and have been registered on ISRCTN registry (https://www.isrctn.com/ISRCTN14800787 registered on 03/06/2025 and https://www.isrctn.com/ISRCTN14174468 registered on 04/06/2025). The study is being conducted by academics and clinicians with expertise in gynaecological oncology (EM), liquid biopsies and genomic analysis (JS/RA/CC/MW), pathology (ES), radiology (YG/ID) and health equity (ND). The study is sponsored by the University of Leicester (protocol numbers 1040 (DOORS-M), 1041 (DOORS-D) and opened for recruitment in July 2025. The clinical research teams will be notified of protocol modifications through the sponsor's research governance team. The research team (EM/JS/CC/MW/ND) are employees of the study sponsor and have designed of the study and will conduct, analyse, and report the outcomes. Given the nature of the studies a Data Monitoring Committee will not be required, instead representatives from the sponsor's research governance team will monitor the conduct of the study and ensure that all procedures and requirements are adhered to. Participant recruitment will continue until 31/01/2028 and data collection will be completed by 28/03/2028 (end of study), with additional follow-up data on patient outcomes

collected up to 2 years following the end of the study. It is anticipated that the results of the qualitative aspects of the studies will be reported in 2026, whereas results from the clinical aspect of the studies will be reported in 2028.

## Public and patient involvement (PPI)

PPI groups and outreach events have been held with women from diverse ethnic backgrounds, including of Black ethnicity, to discuss the challenge of uterine fibroid management and the need to identify uterine sarcomas in a timely manner. The PPI group reported awareness of uterine fibroids but not sarcomas, and were supportive of the development of a blood test to guide diagnosis and management. The study design is similar to a previous cohort study developing a blood test to diagnose endometrial cancer recurrence, which was deemed a very acceptable test by the study participants [35]. A PPI group to advise the research team during the conduct of the study will be recruited.

## Study design

DOORS-D is a single centre case series study that is being conducted in an academic gynaecological oncology centre at the University Hospitals of Leicester. DOORS-D aims to investigate whether genomic alterations in plasma are able to distinguish between uterine sarcomas and uterine fibroids, with a secondary aim to investigate the potential of radiomics and large language models to diagnose sarcomas.

DOORS-M is a cohort study, collecting data and longitudinal samples from patients who have been diagnosed with a uterine sarcoma that aims to create genomic and methylation profiles for uterine sarcomas to identify signatures that could be used within a ctDNA 'test' to detect or predict recurrence. The lead recruitment site will be the University Hospitals of Leicester, Additional recruitment sites will be established however, no blood samples will be collected from the other sites, instead information on patient characteristics, archived tumour samples and sequencing results, and imaging will be collected.

Both studies include secondary objectives of exploring patients' opinions and the psychological impact of ctDNA to diagnose/monitor uterine sarcomas. Table 1 includes the study objectives for the two studies.

## Sample size

DOORS-D and DOORS-M both aim to recruit 50 patients. Sample size calculations for DOORS-D have determined that for a cohort of 50 patients a 40% prevalence of uterine sarcoma (20 cases) would give a 95% confidence interval of width

**Table 1. Objectives for DOORS-D and DOORS-M studies.**

| **DOORS-D** | |
|---|---|
| Primary objective | To investigate whether genomic alterations in plasma are able to distinguish between uterine sarcomas and fibroids |
| Secondary objectives | To investigate the ability of a ctDNA 'test' to detect uterine sarcomas<br>To investigate the potential of Artificial Intelligence (AI) to diagnose uterine sarcomas<br>To explore the psychological impact of ctDNA testing to guide uterine sarcoma diagnosis |
| **DOORS-M** | |
| Primary objective: | To create genomic and methylation profiles for uterine sarcomas |
| Secondary objectives: | To investigate the ability of a ctDNA 'test' to detect uterine sarcoma recurrence<br>To investigate the potential of AI to diagnose uterine sarcomas<br>To explore the psychological impact of ctDNA testing in uterine sarcomas follow-up<br>To identify a symptom profile associated with uterine sarcomas |

0.247 for a sensitivity of 0.95 (95% CI: 0.751–0.999). DOORS-M aims to recruit 50 patients with a previous diagnosis of uterine sarcoma. Given the high rate of uterine sarcoma recurrence, recruitment will be achievable within the study duration, given that the University Hospitals of Leicester currently discusses more than 50 patients with 'suspicious looking fibroid, sarcoma cannot be excluded' in the multidisciplinary team meetings per year of which 7–10 are confirmed to be a sarcoma on histology examination.

## Study population

DOORS-D will recruit women due to undergo surgery, either hysterectomy or myomectomy, for a suspected uterine sarcoma or a fibroid, whereas DOORS-M will recruit patients who have already received a uterine sarcoma diagnosis within the previous 10 years. Adult females aged 18–99 years, who meet the above criteria, are willing and able to consent to participate in the trial, and understand study requirements, with interpreter support if needed, will be offered recruitment.

## Consent procedures

Potential participants will be given a participant information sheet, the opportunity to ask questions, and be given at least 24 hours decision time before consent is taken. Written consent will be taken by a member of the clinical research team and will include a discussion on the aims and objectives of the study, the study processes and potential risks. It will be explained to the potential participant that they are free to withdraw at any time, they are not obliged to give a reason, and that this will not impact on their future care. Women who do not have English as their primary language will be offered an interpreter to support the consent process.

## Study processes – DOORS-D

The schedule of enrolment, interventions and assessment for DOORS-D are described in Fig 1 [36].

Participants recruited to DOORS-D study will have a Case Report Form (CRF) completed detailing their demographic characteristics, medical history, and current clinical episode. Pre-operative radiological imaging, including ultrasound scans, CT/MR/PET scans will undergo a blinded review by the study radiologist, and representative images selected for future AI analysis. The analysis plan using radiomics and large language models has not been determined *a priori* due

| | TRIAL PERIOD | | | |
|---|---|---|---|---|
| | Enrollment | | Pre-operative | Post-operative |
| TIMEPOINT | *-10 weeks to 0* | *0* | *± 10 weeks* | *± 10 weeks* |
| ENROLLMENT | | X | | |
| Eligibility screen | X | | | |
| Informed consent | X | | | |
| ASSESSMENTS: | | | X | X |
| *CRF completion* | | | X | |
| *Blood sample* | | | X | |
| *Interview* | | | | X |

**Fig 1. Participant timeline: Schedule of enrolment, interventions, and assessments for DOORS-D.**

to rapid changes within the field. Prior to surgery 20mls of blood will be collected, processed into plasma and stored for batched ctDNA analysis. The histological diagnosis from surgery will be classified into 1) uterine fibroid; 2) uterine sarcoma; or 3) smooth muscle tumour of uncertain malignant potential (STUMP). A blinded pathological review will be undertaken by the study pathologist to confirm the diagnosis. Participants diagnosed with a sarcoma or STUMP will be offered recruitment to the DOORS-M study. DNA will be extracted from pathology blocks, either fresh or formalin fixed, for whole exome sequencing (WES). Following surgery, a subset of participants will be invited to an interview to discuss their experiences of the study. Recruitment will be purposeful in order to ensure a breadth of opinions from participants from diverse age, socioeconomic and ethnic populations. The interviews will be transcribed verbatim using Microsoft Teams and will be analysed using a reflexive thematic analysis, as described by Braun and Clarke [37].

## Study processes – DOORS-M

The schedule of enrolment, interventions and assessment for DOORS-M are described in Fig 2.

Potential participants will be identified through their clinical care team, screened, and if eligible approached by a member of the research team for recruitment. Once recruited, information will be collected on a CRF to collate information on the patient's demographics, previous medical history and sarcoma diagnosis/treatment. Tumour blocks from previous diagnostic or treatment biopsies will be requested for DNA extraction and WES, unless previously sequenced as part of the patient's clinical care. A blinded pathological review will be undertaken by the study pathologist to confirm the diagnosis. Pre-operative imaging will undergo a blinded review following the same process outlined in DOORS-D, and representative images stored for future AI analysis. At recruitment the participant will have 20mls of blood collected for processing into plasma and storage for ctDNA analysis. Given the constant advances in the field we have not predetermined *a priori* the ctDNA analysis technique. Further blood samples will be collected at subsequent clinic appointments, or 3–6 monthly, for as long as the patient remains within the study. Only participants recruited at the University Hospitals of Leicester will undergo blood sampling due to logistic and financial constraints however, participants will consent to analysis of archival tumour tissue for DNA extraction, thereby expanding the number of sarcoma cases available for analysis. Recruitment from other sites will also involve CRF completion and participant interviews, thereby extending the geographical spread of the study. As with DOORS-D, a subset of participants will be recruited for a semi-structured qualitative interview to discuss

| | STUDY PERIOD | | | | | | |
|---|---|---|---|---|---|---|---|
| | Enrollment | | Post-recruitment | | | | |
| **TIMEPOINT** | *-10 weeks* to 0 | 0 | ± 10 weeks | ± 10 weeks | ± 10 weeks | ± 10 weeks | ± 10 weeks |
| **ENROLLMENT:** | | X | | | | | |
| Eligibility screen | X | | | | | | |
| Informed consent | X | | | | | | |
| **ASSESSMENTS:** | | | | | | | |
| *CRF completion* | | X | | | | | |
| *Blood sample for ctDNA analysis* | | | X | X | X | X | X |
| *Interview* | | | | | | | X |

**Fig 2. Participant timeline: Schedule of enrollment, interventions, and assessments for DOORS-M study.**

their opinions and the potential psychological implications of using ctDNA to diagnose, and/or predict, sarcoma recurrence. The selection of participants will aim to capture a range of opinions and experiences and include an ethnically and socioeconomically diverse population.

### Safety considerations

The DOORS-D and DOORS-M studies are data/sample collection studies and the only physical study procedure participants will be undertaking is venepuncture. The qualitative interview may bring up upsetting memories and participants will be advised that they may choose not to answer a question if they do not wish to. The research team undertaking the interviews are experienced in conducting interviews with oncology populations, following a distress protocol when indicated, supporting participants, and signposting to additional support in order to minimise any distress, including clinical nurse specialists and cancer charity support. Should analysis of a participant's tumour, radiology, and/or ctDNA result in the identification of an abnormality or genetic mutation that could impact the patient, or their family members, for example BRCA mutations or Lynch syndrome, this result will be communicated to the patient's clinical care team for onward investigation and if advised further investigations or referrals.

### Analysis

Data from the two studies will be collected from all the participants using the CRFs and descriptive statistics used to report patient demographics and tumour subtype characteristics. Missing data will be minimised by making efforts to ensure complete follow-up. Given the study design all available data will be included and participants will be censored at their last clinic appointment. Correlation between the second pathology and radiology reviews and the patient's original diagnosis will be undertaken (kappa coefficient). Genomic and/or methylation profiles generated from the study data will be descriptive. Descriptive statistics will be used to report the sensitivity, specificity and diagnostic accuracy of ctDNA and large language model analysis of radiological images to differentiate uterine sarcomas from uterine fibroids. The qualitive interviews will be analysed using a reflexive thematic analysis, as described by Braun and Clarke [37].

### Study timelines

The studies opened at the University Hospitals of Leicester on 14/07/2025 and will continue recruiting until 29 February 2028. Results from the qualitative study will be expected after 1 year of recruitment, whereas the results from the AI analysis of radiological imaging and genomic analysis will be expected following completion of recruitment.

### Discussion

The DOORS-D and DOORS-M studies aim to explore the potential of a ctDNA-based approach to uterine sarcoma management and have the potential to give greater insights into the landscape and challenges of differentiating uterine sarcomas from uterine fibroids. Each study aims to recruit 50 patients, to attain 20 sarcoma cases (40% prevalence) in DOORS-D and 50 cases in DOORS-M. Although this is a large number of cases for a rare malignancy, there is the possibility that our cohort may not include rarer subtypes. The developed ctDNA test through these studies would be trialled in a larger uterine sarcoma population, thereby determining diagnostic ability and enabling test modification. Such an approach has been explored previously [24] however, as yet there is no test that can be used in everyday clinical practice to guide patient management. Also, there is a need for greater understanding of the genomic landscape of sarcomas from women of Black ethnicity to ensure that any test that is developed has equitable diagnostic accuracy, unlike other tests currently used to diagnose endometrial cancer, for example the transvaginal ultrasound scan which has a lower diagnostic ability in women of Black ethnicity [38]. The studies include qualitative interviews to ensure that patients' voices and opinions are heard and are able to contribute to the future development of blood test diagnosis or monitoring for uterine sarcomas. In addition, exploration of opinions and experiences of women from ethnic minority populations is essential in

determining the potential impact for such a 'test' due to the significantly poorer long-term uterine cancer survival rates [39] and greater prevalence of uterine fibroids [7]. Previous work that the research team has undertaken, identified a lack of awareness of uterine malignancy amongst women from Black ethnic populations resident in England [40] and although efforts are being undertaken to raise awareness of the signs and symptoms of uterine cancer [41, 42], these focus typically on the symptom profile associated with endometrial cancer rather than uterine sarcomas.

There is a clear need to avoid undertreatment and potentially disseminating surgical procedures, however, this needs to be balanced with patients' wishes for fertility preservation, especially in younger women [11]. One of the main challenges is that over half of patients diagnosed with an unexpected uterine sarcoma are pre-menopausal and/or under the age of 50 years, and the common presenting symptoms, irregular uterine bleeding, pelvic mass and menorrhagia, are indistinguishable from symptoms associated with uterine fibroids [5]. A percutaneous core needle biopsy is routinely performed to diagnose suspected soft-tissue sarcomas [43] however, this is an invasive procedure and although the reported dissemination rate is low for retroperitoneal sarcomas [44] there is no data for uterine sarcomas. It also would require a suspicion of sarcoma and therefore may not impact unexpected cases. It also requires clinicians with the relevant skills to perform the procedure, which is unlikely to be available at every hospital, thereby requiring the patient to travel to a centre with the necessary expertise. The potential of a ctDNA test that would negate the need for a biopsy and could reduce the duration of the patient's diagnostic pathway. In addition, genomic information on an individual's sarcoma could potentially be used to guide their management, particularly for immunotherapy, since a number of uterine sarcomas have been identified as having microsatellite instability (MSI) or deletions of *PDCD1*, which encodes PD-1, as well as other actionable mutations such as in *BRCA2* (5%), *BRAF* and *ALK* fusions [22, 45].

Although features associated with uterine sarcomas have been described radiologically [46], the accuracy of differentiating sarcomas from degenerating uterine fibroids using morphological features on MRI is variable, even by expert radiologists [47], sensitivity 70.7% and specificity 76.2% [20]. Radiomic analysis of radiological images may improve diagnostic accuracy with imaging [48, 49] and help reduce intra-observer variability.

In conclusion, the DOORS-D and DOORS-M studies aim to add to the genomic data on uterine sarcomas from a diverse ethnic population, the challenge of correctly differentiating uterine sarcomas from uterine fibroids, and exploring the experiences and opinions of patients who have undergone surgery for a suspected or are confirmed to have a uterine sarcoma.

## Supporting information
**S1 Checklist. SPIRIT checklist.**
(DOCX)

## Acknowledgments
We would like to thank the patients and public who supported the development of this study.

## Author contributions
**Conceptualization:** Jacqui Shaw, Natalie Darko, Esther L Moss.

**Methodology:** Rebecca Allsopp, Jacqui Shaw, Caroline Cowley, Elizabeth Stannard, Yvette Griffin, Indrajeet Das, Marc Wadsley, Esther L Moss.

**Writing – original draft:** Anna Casey, Esther L Moss.

**Writing – review & editing:** Rebecca Allsopp, Jacqui Shaw, Caroline Cowley, Elizabeth Stannard, Yvette Griffin, Indrajeet Das, Marc Wadsley, Natalie Darko, Esther L Moss.

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
