## [Decision Letter · Decision Letter 0]

10 Nov 2025

Dear Dr. Moss,

Thank you for submitting your manuscript to PLOS ONE. After careful consideration, we feel that it has merit but does not fully meet PLOS ONE’s publication criteria as it currently stands. Therefore, we invite you to submit a revised version of the manuscript that addresses the points raised during the review process.

We look forward to receiving your revised manuscript.

Kind regards,

Mohamed Gouda, MD, PhD

Academic Editor

PLOS ONE

Journal Requirements:

“The studies have been funded by a Rare Gynaecological Cancer Research grant (EVE 0042) from The Eve Appeal gynaecological cancer charity. The funder has not had a role in the design of the study and will not be involved in the conduct, analysis, or reporting of trial.”

“EM, JS and DD were the recipients of the grant from The Eve Appeal, the gynaecological cancers charity (EVE 0042) https://eveappeal.org.uk/. The University of Leicester is the study sponsor. The research team (EM/JS and DD) are employees of the sponsor and have designed of the study and will conduct, analyse, and report of outcomes.”

“EM/DD have received research funding from Sarcoma UK for an unrelated sarcoma-focused research study.”

Additional Editor Comments:

Reviewer #1:

This manuscript describes a study protocol aiming to use circulating tumor DNA (ctDNA), a minimally invasive procedure, for diagnosis and monitoring of uterine sarcoma. I have some comments:

- I suggest revising the word order In the title, so that “diagnosis” precedes “monitoring” fitting the sequence of the abbreviations in the title (“DOORS-D and DOORS-M”) and the study objectives.

- Based on the described proposed cohort of 50 patients, 20 will have uterine sarcomas and the remaining 30 will have uterine fibroids. My concern is that leiomyomas are considerably more common and with these small numbers, the study may be underpowered to detect distinct genetic alterations or provide meaningful comparative data between the two categories. Also, limiting sarcoma cases to 20—most likely leiomyosarcomas—may not adequately represent different histotypes.

- Although there are still limitations and variations in MRI performance reported in the literature, the discussion in the manuscript seems to understate its diagnostic value. A meta-analysis of eight studies involving 2,495 women (2,253 with leiomyomas and 179 with uterine sarcomas) reported a pooled sensitivity of 0.90 and specificity of 0.96 for differentiating sarcomas from fibroids (PMID: 37732472). In addition, there is growing consensus on refining MRI evaluation in uterine masses for risk of leiomyosarcoma (PMID: 36194109).

- I would suggest the authors to provide more details on previous efforts exploring the role of ctDNA in uterine sarcomas.

- It would be helpful to discuss the diagnostic challenges during intraoperative pathology assessment, which are often limited by time constraints and restricted tissue sampling. These factors frequently delay definitive diagnosis until permanent sections are reviewed. The proposed study could significantly help bypassing this dilemma.

Reviewer #2:

I think it will be a useful study to answer your scientific question and proof the hypothesis.

However, none is specified about the technique of ctDNA which is core in this particular work.

Reviewer #3:

The manuscript Study Protocol: Development of a blood test for uterine sarcoma – monitoring and diagnosis studies which aims to explore the role of circulating tumor DNA to diagnose and to monitor uterine sarcomas.

The study is sponsored by the University of Leicester and a grant from the Eve Appeal (EVE 0042) and is approved by the South Central – Berkshire B Research Ethics Committee.

Introduction – well written and organized. The study has a strong rationale and potential impact in the diagnosis and care of patients with uterine sarcomas.

Sample size – aim to recruit 50 patients in each study, per the authors it’s feasible based on high volume at the University Hospitals of Leicester.

Study population/Inclusion criteria – DOORS-D women due for hysterectomy/myomectomy for uterine fibroids. DOORS-M previous diagnosis of uterine sarcoma in the previous 10 years. 18-99 years.

Overall, this is a very well written protocol and well-designed studies that answer an important and clinically relevant question. However, there are some comments that could further addressed by the authors:

Methods sections

- authors will use WES as the method to analyze ctDNA from patients diagnosed with uterine sarcomas and will create a tumor-uninformed (agnostic) liquid biopsy test using genomic/methylation profile from those patients to detect uterine sarcoma signatures in patients presenting with fibroids. The authors could provide additional details in this section regarding their analysis and bioinformatics plan.

- There is also no clear guidance on the “semi-structured qualitative interviews” and how those will be evaluated and included in the study.

- There is a mention the use of AI and without providing additional details on the methods planned for this analysis.

Study Design

– The authors mention the inclusion of additional sites without blood collection. The authors could clarify the reason/criteria to include, and which additional sites are planned to be part of the study. In addition, since there is no blood collection, is there a limit of patients from the planned 50 included that will be enrolled in the “additional sites” vs. the University of Leicester?

Primary and secondary objectives

– the authors describe a very broad definition for primary and secondary objectives, without providing a clear outcome/value to consider the study positive. Based on sample size calculation they expect 40% prevalence of uterine sarcoma, what is the accuracy, sensitivity/specificity and ROC/AUC to consider this test feasible to use in clinical practice?

Reviewers' comments:

Reviewer's Responses to Questions

**Comments to the Author**

1. Does the manuscript provide a valid rationale for the proposed study, with clearly identified and justified research questions?

Reviewer #1: Yes

Reviewer #2: Yes

Reviewer #3: Yes

2. Is the protocol technically sound and planned in a manner that will lead to a meaningful outcome and allow testing the stated hypotheses?

Reviewer #1: Yes

Reviewer #2: Partly

Reviewer #3: Yes

3. Is the methodology feasible and described in sufficient detail to allow the work to be replicable?

Reviewer #1: Yes

Reviewer #2: No

Reviewer #3: No

4. Have the authors described where all data underlying the findings will be made available when the study is complete?

Reviewer #1: Yes

Reviewer #2: Yes

Reviewer #3: Yes

5. Is the manuscript presented in an intelligible fashion and written in standard English?

Reviewer #1: Yes

Reviewer #2: Yes

Reviewer #3: Yes

You may also provide optional suggestions and comments to authors that they might find helpful in planning their study.

Reviewer #1: This manuscript describes a study protocol aiming to use circulating tumor DNA (ctDNA), a minimally invasive procedure, for diagnosis and monitoring of uterine sarcoma. I have some comments:

- I suggest revising the word order In the title, so that “diagnosis” precedes “monitoring” fitting the sequence of the abbreviations in the title (“DOORS-D and DOORS-M”) and the study objectives.

- Based on the described proposed cohort of 50 patients, 20 will have uterine sarcomas and the remaining 30 will have uterine fibroids. My concern is that leiomyomas are considerably more common and with these small numbers, the study may be underpowered to detect distinct genetic alterations or provide meaningful comparative data between the two categories. Also, limiting sarcoma cases to 20—most likely leiomyosarcomas—may not adequately represent different histotypes.

- Although there are still limitations and variations in MRI performance reported in the literature, the discussion in the manuscript seems to understate its diagnostic value. A meta-analysis of eight studies involving 2,495 women (2,253 with leiomyomas and 179 with uterine sarcomas) reported a pooled sensitivity of 0.90 and specificity of 0.96 for differentiating sarcomas from fibroids (PMID: 37732472). In addition, there is growing consensus on refining MRI evaluation in uterine masses for risk of leiomyosarcoma (PMID: 36194109).

- I would suggest the authors to provide more details on previous efforts exploring the role of ctDNA in uterine sarcomas.

- It would be helpful to discuss the diagnostic challenges during intraoperative pathology assessment, which are often limited by time constraints and restricted tissue sampling. These factors frequently delay definitive diagnosis until permanent sections are reviewed. The proposed study could significantly help bypassing this dilemma.

Reviewer #2: I think it will be a useful study to answer your scientific question and proof the hypothesis.

However, none is specified about the technique of ctDNA which is core in this particular work.

Reviewer #3: The manuscript Study Protocol: Development of a blood test for uterine sarcoma – monitoring and diagnosis studies which aims to explore the role of circulating tumor DNA to diagnose and to monitor uterine sarcomas.

The study is sponsored by the University of Leicester and a grant from the Eve Appeal (EVE 0042) and is approved by the South Central – Berkshire B Research Ethics Committee.

Introduction – well written and organized. The study has a strong rationale and potential impact in the diagnosis and care of patients with uterine sarcomas.

Sample size – aim to recruit 50 patients in each study, per the authors it’s feasible based on high volume at the University Hospitals of Leicester.

Study population/Inclusion criteria – DOORS-D women due for hysterectomy/myomectomy for uterine fibroids. DOORS-M previous diagnosis of uterine sarcoma in the previous 10 years. 18-99 years.

Overall, this is a very well written protocol and well-designed studies that answer an important and clinically relevant question. However, there are some comments that could further addressed by the authors:

Methods sections

- authors will use WES as the method to analyze ctDNA from patients diagnosed with uterine sarcomas and will create a tumor-uninformed (agnostic) liquid biopsy test using genomic/methylation profile from those patients to detect uterine sarcoma signatures in patients presenting with fibroids. The authors could provide additional details in this section regarding their analysis and bioinformatics plan.

- There is also no clear guidance on the “semi-structured qualitative interviews” and how those will be evaluated and included in the study.

- There is a mention the use of AI and without providing additional details on the methods planned for this analysis.

Study Design

– The authors mention the inclusion of additional sites without blood collection. The authors could clarify the reason/criteria to include, and which additional sites are planned to be part of the study. In addition, since there is no blood collection, is there a limit of patients from the planned 50 included that will be enrolled in the “additional sites” vs. the University of Leicester?

Primary and secondary objectives

– the authors describe a very broad definition for primary and secondary objectives, without providing a clear outcome/value to consider the study positive. Based on sample size calculation they expect 40% prevalence of uterine sarcoma, what is the accuracy, sensitivity/specificity and ROC/AUC to consider this test feasible to use in clinical practice?

**Do you want your identity to be public for this peer review?** For information about this choice, including consent withdrawal, please see our Privacy Policy

Reviewer #1: No

Reviewer #2: **Yes:** Oriol Mirallas

Reviewer #3: **Yes:** Eduardo Edelman Saul

---

## [Author Response · Author response to Decision Letter 1]

21 Nov 2025

Dear Editor,

thank you for the opportunity to respond to the reviewers’ comments on your manuscript. I have responded point by point below, and will wait to hear from you,

Kind regards

Esther

Thank you. We have made the style changes.

“The studies have been funded by a Rare Gynaecological Cancer Research grant (EVE 0042) from The Eve Appeal gynaecological cancer charity. The funder has not had a role in the design of the study and will not be involved in the conduct, analysis, or reporting of trial.”

“EM, JS and DD were the recipients of the grant from The Eve Appeal, the gynaecological cancers charity (EVE 0042) https://eveappeal.org.uk/. The University of Leicester is the study sponsor. The research team (EM/JS and DD) are employees of the sponsor and have designed of the study and will conduct, analyse, and report of outcomes.”

Thank you. The updated funding statement is

‘EM, JS and DD were the recipients of the grant from The Eve Appeal, the gynaecological cancers charity (EVE 0042) https://eveappeal.org.uk/. The University of Leicester is the study sponsor. The research team (EM/JS and DD) are employees of the sponsor and have designed of the study and will conduct, analyse, and report of outcomes. This study will be delivered through the National Institute for Health and Care Research (NIHR) Leicester Biomedical Research Centre. The views expressed are those of the author(s) and not necessarily those of The Eve Appeal, the NIHR Leicester BRC, the NIHR or the Department of Health and Social Care.’ I have uploaded an amended cover letter with the advised changes.

“EM/DD have received research funding from Sarcoma UK for an unrelated sarcoma-focused research study.”

Thank you. This has been added to the cover letter.

The supporting information of the Spirit checklist and study protocol have been submitted as supporting information for the publication process however we did not intend for them to be included in the publication.

Additional Editor Comments:

Reviewer #1:

This manuscript describes a study protocol aiming to use circulating tumor DNA (ctDNA), a minimally invasive procedure, for diagnosis and monitoring of uterine sarcoma. I have some comments:

- I suggest revising the word order In the title, so that “diagnosis” precedes “monitoring” fitting the sequence of the abbreviations in the title (“DOORS-D and DOORS-M”) and the study objectives.

This has been amended.

- Based on the described proposed cohort of 50 patients, 20 will have uterine sarcomas and the remaining 30 will have uterine fibroids. My concern is that leiomyomas are considerably more common and with these small numbers, the study may be underpowered to detect distinct genetic alterations or provide meaningful comparative data between the two categories. Also, limiting sarcoma cases to 20—most likely leiomyosarcomas—may not adequately represent different histotypes.

Thank you for raising this important point. The aim is to recruit 50 patients for each study and therefore we aim to recruit a minimum of 70 sarcoma cases (DOORS-D 20 cases, DOORS-M 50 cases). We that this is still not a large number of cases and may not include all the potential subtypes given the rarer subtypes. The developed ctDNA test would then be trialled in a larger sarcoma population, which would enable its diagnostic ability to be determined. Additional text has been added to the discussion on this important point (line 296-301).

‘Each study aims to recruit 50 patients for each study, 20 sarcoma cases in DOORS-D and 50 cases in DOORS-M. Although this is a large number of cases for a rare malignancy, there is the possibility that our cohort may not include rarer subtypes. The developed ctDNA test through these studies would be trialled in a larger uterine sarcoma population, thereby determining diagnostic ability and enabling test modification.’

- Although there are still limitations and variations in MRI performance reported in the literature, the discussion in the manuscript seems to understate its diagnostic value. A meta-analysis of eight studies involving 2,495 women (2,253 with leiomyomas and 179 with uterine sarcomas) reported a pooled sensitivity of 0.90 and specificity of 0.96 for differentiating sarcomas from fibroids (PMID: 37732472). In addition, there is growing consensus on refining MRI evaluation in uterine masses for risk of leiomyosarcoma (PMID: 36194109).

Thank you for your comment. The section on imaging has been expanded in the introduction and these two references included:

Imaging, in particular MRI, can be used to differentiate between uterine sarcomas and fibroids with a high level of diagnostic accuracy [15]. Features including peritoneal or lymph node metastases, high/intermediate on T2-weighted imaging, low apparent diffusion coefficient (ADC) and high intensity signal of diffusion weighted images were identified as having a strong association with uterine sarcomas [16]. Whereas other features, such an irregular margin, haemorrhage and enhancement, were determined as being an indeterminate association [16], and it is reported that there is a greater challenge when differentiating degenerating or complex fibroids from uterine sarcomas [17]. Lines 80-89

- I would suggest the authors to provide more details on previous efforts exploring the role of ctDNA in uterine sarcomas.

Thank you. Additional text has been added:

The potential of the ‘liquid biopsy’ toto monitor uterine sarcomas has been explored in previous studies where ctDNA has shown promise as a marker both in the monitoring [30-32] or diagnosis [23] of uterine sarcomas, although other analytes including microRNAs have also been shown to have potential as biomarkers [33]. Leiomyosarcomas were the most common histological subtype investigated and there was significant heterogeneity in terms of both laboratory approaches and the reporting of these approaches. Given the small number of uterine sarcoma cases in the majority of the studies to date, the proportion of cases with detectable ctDNA is currently unknown, and there is no cost-effective test that could give a result in a short time frame to guide patient management, thereby reducing pre-operative diagnostic uncertainty. Line 111-120

- It would be helpful to discuss the diagnostic challenges during intraoperative pathology assessment, which are often limited by time constraints and restricted tissue sampling. These factors frequently delay definitive diagnosis until permanent sections are reviewed. The proposed study could significantly help bypassing this dilemma.

Thank you for raising this issue. Additional text has been added on this point:

Intraoperative pathology assessment can help with guiding patient management however, this is can be associated with diagnostic challenges [9], and appears to be less concordant with final histology compared to assessment of other gynaecological malignancies [10]. Line 69-72

Reviewer #2:

I think it will be a useful study to answer your scientific question and proof the hypothesis.

However, none is specified about the technique of ctDNA which is core in this particular work.

Thank you for your comment. Given the constant advances in the field we did not want to specify the technique that will be used at this stage, hence the lack of information on this point. This point has been added to the text.

‘Given the constant advances in the field we did have not predetermined a priori the ctDNA analysis technique.’ Line 224-5

Reviewer #3:

The manuscript Study Protocol: Development of a blood test for uterine sarcoma – monitoring and diagnosis studies which aims to explore the role of circulating tumor DNA to diagnose and to monitor uterine sarcomas.

The study is sponsored by the University of Leicester and a grant from the Eve Appeal (EVE 0042) and is approved by the South Central – Berkshire B Research Ethics Committee.

Introduction – well written and organized. The study has a strong rationale and potential impact in the diagnosis and care of patients with uterine sarcomas.

Thank you

Sample size – aim to recruit 50 patients in each study, per the authors it’s feasible based on high volume at the University Hospitals of Leicester.

Study population/Inclusion criteria – DOORS-D women due for hysterectomy/myomectomy for uterine fibroids. DOORS-M previous diagnosis of uterine sarcoma in the previous 10 years. 18-99 years.

Overall, this is a very well written protocol and well-designed studies that answer an important and clinically relevant question. However, there are some comments that could further addressed by the authors:

Methods sections

- authors will use WES as the method to analyze ctDNA from patients diagnosed with uterine sarcomas and will create a tumor-uninformed (agnostic) liquid biopsy test using genomic/methylation profile from those patients to detect uterine sarcoma signatures in patients presenting with fibroids. The authors could provide additional details in this section regarding their analysis and bioinformatics plan.

As mentioned in the response to reviewer 2, given the constant advances in the field of liquid biopsy and ctDNA we are unable to give more detail at this stage on the analysis plan. Instead, when the samples are available, in approximately 2 years’ time, the ctDNA experts within the team will undertake the most appropriate analysis techniques. This point has been added to the text.

- There is also no clear guidance on the “semi-structured qualitative interviews” and how those will be evaluated and included in the study.

The interview studies have been included to ensure that patients’ voices and opinions are heard and are able to contribute to the future development of blood test diagnosis or monitoring for uterine sarcomas. Additional information on the qualitative research methodology has been included.

‘The interviews will be transcribed verbatim using Microsoft Teams and will be analysed using a reflexive thematic analysis, as described by Braun and Clarke28.’ Line 208-210

‘The studies have been included qualitative interviews to ensure that patients’ voices and opinions are heard and are able to contribute to the future development of blood test diagnosis or monitoring for uterine sarcomas.’ Line 310-312.

- There is a mention the use of AI and without providing additional details on the methods planned for this analysis.

As with the ctDNA analysis, the analysis plan for the use of AI or radiomics has not yet been developed. Additional information has been added to the text.

The AI analysis plan and use of radiomics has not been determined a priori due to rapid changes within the field.’ Line 199-200

Study Design

– The authors mention the inclusion of additional sites without blood collection. The authors could clarify the reason/criteria to include, and which additional sites are planned to be part of the study. In addition, since there is no blood collection, is there a limit of patients from the planned 50 included that will be enrolled in the “additional sites” vs. the University of Leicester?

At this present time there are only resources to collect blood sample at the University of Leicester. Participants recruited from the additional sites will be recruited to grant access to their archival tumour tissue for DNA extraction, thereby expanding the number of sarcoma cases available for analysis. Additional text has been added

‘Only participants recruited at the University Hospitals of Leicester will undergo blood sampling due to logistic and financial constraints however, participants will consent to analysis of archival tumour tissue for DNA extraction, thereby expanding the number of sarcoma cases available for analysis. Recruitment from other sites will also involve CRF completion and participant interviews, thereby extending the geographical spread of the study.’ Line 230-235

Primary and secondary objectives

– the authors describe a very broad definition for primary and secondary objectives, without providing a clear outcome/value to consider the study positive. Based on sample size calculation they expect 40% prevalence of uterine sarcoma, what is the accuracy, sensitivity/specificity and ROC/AUC to consider this test feasible to use in clinical practice?

This is an exploratory study to develop the ‘test’ rather than a diagnostic accuracy test, therefore such objectives have not been included since the study has not been powered for these outcomes. Once we have developed the test in this current study a much larger study will be undertaken in order to determine the accuracy of the ‘test’ and the potential for clinical implementation. More detail on this has been added to the text. Line 306-8

---

## [Editor Report · Decision Letter 1]

7 Dec 2025

Study Protocol: Development of a blOOd test for uteRine Sarcoma – Monitoring and Diagnosis (DOORS-D and DOORS-M) studies

PONE-D-25-49314R1

Dear Dr. Moss,

We’re pleased to inform you that your manuscript has been judged scientifically suitable for publication and will be formally accepted for publication once it meets all outstanding technical requirements.

Kind regards,

Mohamed Gouda, MD, PhD

Academic Editor

PLOS One

---

## [Editor Report · Acceptance letter]

PONE-D-25-49314R1

PLOS One

Dear Dr. Moss,

I'm pleased to inform you that your manuscript has been deemed suitable for publication in PLOS One. Congratulations! Your manuscript is now being handed over to our production team.

Kind regards,

on behalf of

Dr. Mohamed Gouda

Academic Editor

PLOS One